# The Reconstitution Dynamics of Cultivated Hematopoietic Stem Cells and Progenitors Is Independent of Age

**DOI:** 10.3390/ijms23063160

**Published:** 2022-03-15

**Authors:** Frauke Gotzhein, Tim Aranyossy, Lars Thielecke, Tanja Sonntag, Vanessa Thaden, Boris Fehse, Ingo Müller, Ingmar Glauche, Kerstin Cornils

**Affiliations:** 1Clinic of Pediatric Hematology and Oncology, Division of Pediatric Stem Cell Transplantation and Immunology, University Medical Center Hamburg-Eppendorf, 20246 Hamburg, Germany; gotzhein@kinderkrebs-forschung.de (F.G.); thaden@kinderkrebs-forschung.de (V.T.); i.mueller@uke.de (I.M.); 2Research Institute Children’s Cancer Center Hamburg, 20251 Hamburg, Germany; 3Research Department Cell and Gene Therapy, Department of Stem Cell Transplantation, University Medical Center Hamburg-Eppendorf, 20246 Hamburg, Germany; t.aranyossy@gmx.de (T.A.); t.sonntag@uke.de (T.S.); fehse@uke.de (B.F.); 4Institute for Medical Informatics and Biometry, Faculty of Medicine Carl Gustav Carus, Technische Universität Dresden, 01307 Dresden, Germany; lars.thielecke@tu-dresden.de (L.T.); ingmar.glauche@tu-dresden.de (I.G.)

**Keywords:** genetic barcoding, hematopoietic stem cells, hematopoietic progenitor cells, stem cell transplantation, clonality, aging

## Abstract

Hematopoietic stem cell transplantation (HSCT) represents the only curative treatment option for numerous hematologic malignancies. While the influence of donor age and the composition of the graft have already been examined in clinical and preclinical studies, little information is available on the extent to which different hematological subpopulations contribute to the dynamics of the reconstitution process and on whether and how these contributions are altered with age. In a murine model of HSCT, we therefore simultaneously tracked different cultivated and transduced hematopoietic stem and progenitor cell (HSPC) populations using a multicolor-coded barcode system (BC32). We studied a series of age-matched and age-mismatched transplantations and compared the influence of age on the reconstitution dynamics. We show that reconstitution from these cultured and assembled grafts was substantially driven by hematopoietic stem cells (HSCs) and multipotent progenitors (MPPs) independent of age. The reconstitution patterns were polyclonal and stable in all age groups independently of the variability between individual animals, with higher output rates from MPPs than from HSCs. Our experiments suggest that the dynamics of reconstitution and the contribution of cultured and individually transduced HSPC subpopulations are largely independent of age. Our findings support ongoing efforts to expand the application of HSCT in older individuals as a promising strategy to combat hematological diseases, including gene therapy applications.

## 1. Introduction

Since the first successful hematopoietic stem cell transplantation (HSCT) in 1957 [1], this therapy has been established as standard of care for many (non-) malignant blood diseases and often represents the only curative treatment available. In the past, a worldwide increase in the demand for HSCTs was reported, surpassing 50,000 allogeneic transplants annually [2]. Since donor age is one of the most predictive, non-HLA donor characteristics for survival after allogeneic HSCT [3,4], the frequency of donors below 30 years of age raised up to 69% in the past years [5]. Regarding the recipients, multiple studies showed that HSCTs in older patients result in worse outcomes compared to younger patients, particularly due to high rates of transplant-related non-relapse mortality and morbidity caused by traditional myeloablative regimens or infections [6,7,8]. As a result, elderly patients were often excluded from HSCT. Fortunately, with the emergence of less toxic, particularly, reduced-intensity conditioning regimens (RICs), the application of HSCTs as a potential treatment option for increasingly older patients has been reevaluated over the last decades [9,10,11]. Despite ongoing efforts to evaluate the risk associated with donor clonal hematopoiesis [12,13], the questions remain of how the dynamics of hematopoietic recovery after HSCT are affected by the age of the donor or of the recipient and of how this process might be accelerated in order to alleviate infections in the early post-transplant phase.

Similar questions are raised in the context of gene therapeutic applications, in which ex vivo cultured and transduced hematopoietic stem and progenitor cells (HSPCs) are transferred back to the same patient via autologous transplantations. This greatly expanding field will target an increasing number of diseases and also be applicable to older patients, for whom optimal conditioning is required.

Besides the influence of donor and recipient age, the cell source (i.e., bone marrow (BM)), mobilized peripheral blood stem cells, and umbilical cord blood) and the composition of the graft have an impact on the success of HSCTs. The surface marker CD34 is used to enrich for HSCs; however, the CD34 protein is expressed not only on HSCs but also on a variety of different progenitor cells [14,15]. While it is well recognized that HSCs confer a long-term repopulation potential, it is less known how different hematopoietic progenitor subpopulations simultaneously contribute to hematopoietic recovery after transplantation [16]. Furthermore, the optimization of the culturing and the transduction processes bears the potential to further improve the success of autologous transplantations.

In recent years, the development of genetic barcodes as a means of cellular labeling has revolutionized clonal tracking [17,18], especially in the field of hematopoiesis [19]. While newer methods for in vitro barcoding allow studying unperturbed hematopoiesis [20,21,22,23], established approaches use integrating retroviral vectors equipped with artificial genetic sequences, termed barcodes, which are introduced in the genome of the target cells and remain there as a cell-specific and permanent marker. As the same retroviral transfer methods are used in gene therapy to inheritably integrate a corrected gene, ex vivo cellular barcoding is the method of choice to follow the clonal progeny of hematopoietic cells after transplantation. Since the genetic barcodes are inheritable, the productivity of the initially marked cells during and after reconstitution can be measured by the abundance of the respective label within the hematopoietic organs (e.g., bone marrow, spleen) as well as in sorted mature blood subpopulations.

The majority of the published work on clonal barcoding utilized highly purified murine HSC populations, sorted according to their respective marker profiles [24,25,26,27]. Other longitudinal studies of blood reconstitution, working with humanized mouse or primate models, investigated grafts consisting of a mixed CD34^+^ cell population, similarly to a standard human transplantation setting [28,29,30,31,32,33,34,35,36]. The difference in clonal composition and clonal contribution between highly purified, aged, and young HSCs was so far only studied by Verovskaya et al. in a murine model utilizing competitive transplantations [24].

We recently developed a novel approach to inheritably mark individual cells, which combines the virtues of two formerly established marking techniques, namely, red-green-blue (RGB) marking [37] and the aforementioned genetic barcoding. While fluorescence-based RGB marking allows for a phenotypic distinction of different cell clones in situ, genetic marking facilitates their robust and long-term follow-up even if the expression of the fluorescence genes is diminished or extinct. This combination of techniques enables the simultaneous analysis of four different cell populations with up to 4^32^ uniquely labeled clones each [38,39]. In the past, we successfully utilized this approach to investigate different viral vectors and the influence of their respective promoters on the reconstitution of peripheral blood [40].

In this study, we compared how donor and recipient age influence the differential contribution of cultured, individually transduced, and simultaneously transplanted hematopoietic subpopulations and drive their reconstitution after HSCT. To this end, we applied our lentiviral marking approach to individually label four different BM subpopulations and to further quantify their distinct contributions on a clonal level. We particularly focused on the contribution of cells from matched young (young into young, Y-Y), aged (old into old; O-O), and one age-mismatched (young into old; Y-O) transplantation settings. Subpopulations of the prepared grafts were chosen to mimic the level of heterogeneity which is usually observed in human grafts for HSCT. Our results provide insights into the age-dependency of the reconstitution processes after HSCT.

## 2. Results

The schematic representation of the lentiviral vectors, including the genetic barcodes as well as the experimental set-up, is provided in Appendix B (Figure A1).

### 2.1. Graft Production and Quality Assessment

In order to study the influence of donor and recipient age on the reconstitution kinetics of different transduced hematopoietic subpopulations, we worked on three different experimental transplantation settings: we transplanted cell populations from young donors into young recipients (Y-Y), from young donors into old recipients (Y-O), and from old donors into old recipients (O-O) (Figure 1A). Technically, the BM of, in total, 30 young or old male donor mice was used for the enrichment of lineage-negative cells, followed by subsequent fluorescence-activated cell sorting (FACS) to further separate HSCs, MPPs, common myeloid progenitors (CMPs), and common lymphoid progenitors (CLPs) using the lineage-negative fraction and staining for the surface expression of Sca-1, cKit, CD34, and CD150 (Appendix B Figure A1 and Appendix A). On average, we obtained similar cell numbers of MPPs and CMPs from old or young donors per animal, while the number of HSCs was almost two times higher after sorting from the aged donor mice (Figure 1B). Additionally, the CLP number in the Y-Y setting was more than two times higher compared to that in the other experimental settings.

Next, the FACS-sorted populations were cultured and transduced individually with one of the four barcoded lentiviral vector libraries (Appendix B Figure A1), with low transduction rates to ensure single vector copy numbers [41]. Each lentiviral construct contained one of four identifying fluorescent proteins (FP) and a unique genetic barcode to discriminate (i) populations by flow cytometry and (ii) clones by their inherited unique barcode sequence. As depicted in Appendix B Figure A1, fixed base triplets within the barcode sequence vary between the different libraries to additionally cipher the FP [38,40]. In order to minimize the potential backbone-related systematic bias, we permutated the viral vector libraries between the experimental settings (Figure 1B).

While in the Y-Y experimental setting we measured moderate transduction rates of 28.9% (21.5–42.6%), the transduction efficiencies in the other two settings exceeded the target range, reaching levels of 71.5% (48.6–94.0%) (Figure 1C), for which multiple integrations per cell become more likely. For the Y-O HSCs, we could not measure the transduction efficiency via flow cytometry due to low cell numbers. Instead, we used the vector copy number (VCN) obtained from single-cell analysis of spleen cells from the transplanted animals and fitted a Poisson distribution to estimate the overall transduction efficiency (Figure 1C and Appendix A). Taken together, we successfully separated by FACS and transduced four HSPC populations from the BM of young and aged male donor mice.

### 2.2. Dynamics of Blood Reconstitution from Old and Young Grafts

After pooling the transduced HSPCs and mixing with the BM support from female mice age-matched to the recipients, we transplanted the resulting graft into 30 lethally irradiated female recipients. To avoid any loss of cells, we refrained from additional cell counting before mixing. Proportions of the HPSCs per animals at the time of sorting were 0.86–1.62% for HSCs, 1.67–2.04% for MPPs, 7.64–8.89% for CMPs, and 0.92–2.15% for CLPs, exemplarily calculated for one graft. To follow the engraftment of the donor cells, we performed digital droplet PCR (ddPCR) on the Y chromosome to determine male/female chimerism in the BM after transplantation [40,42]. We saw a continuous increase in transplanted male cells in the bone marrow of the recipients over time in all settings, achieving chimerism levels of 68% in the Y-Y experiment, 44% in the Y-O experiment, and 56% in the O-O experiment after 16 weeks (Figure 2A). Although we detected some statistically significant differences in the mean chimerism level between the groups, there was no consistent trend rendering one experimental group clearly superior to the others. Especially at the end of the observation period, comparable chimerism levels were achieved. Moreover, the pronounced variability within each group (covering ranges up to 40%) repeatedly exceeded these differences in the means and did not support the notion of a relevant difference in the reconstitution dynamics. We concluded that the reconstitution dynamics were largely comparable between the three experimental groups.

To monitor the overall reconstitution of different blood cell subpopulations, we performed flow cytometry on single-cell suspensions from the spleens of the transplanted animals. Specifically, we assessed T cells (stained by CD3e) and B cells (stained by B220) from the lymphoid compartment and monocytes/macrophages (stained by CD11b) and granulocytes (stained by CD11b and Ly6G) from the myeloid compartment. As shown in Figure 2B, the relative number of donor-derived leukocytes increased for three weeks post transplantation and then stabilized. No significant differences were observed in the reconstitution dynamics of different leukocyte subsets, pointing towards an independence of the age of the donor material (Figure 2C). B cells expanded rapidly and constantly, whereas the other subpopulations remained stable during the observation time of 16 weeks. By comparing the overall reconstitution kinetics, we did not observe pronounced differences between all three groups, indicating no or only a weak influence of age on the reconstitution dynamics of cultured and transduced cell populations.

### 2.3. Contribution of the Transduced Cell Populations to Hematopoietic Reconstitution

To compare how the transduced stem and progenitor cells differentially contributed to hematopoietic reconstitution, we assessed their relative abundance and the abundance of their progeny in the spleen by quantifying the expression of the respective fluorescent proteins (Figure 3). In the Y-Y experimental setting, the overall contribution was found to be extremely low, as FP expression values mostly ranged below 2%. An exception was measured at week 3, in which we detected an extraordinarily high contribution of MPP-derived cells in all mice. Sixteen weeks after transplantation, the expression of fluorescence markers declined to nearly zero (Figure 3A). On the contrary, in old recipients (Figure 3B,C), we detected a higher number of transduced cells. In both groups, the contribution of transduced HSCs and MPPs peaked at around 10% at the later time points. Within the Y-O setting, we observed an almost stable MPP contribution during the entire observation time. This was not present in the O-O experiment, where the MPP-derived progeny declined after a peak at week 3. In general, compared to HSC-derived cells, the MPP output was consistently higher at all time points, independent of donor or recipient age. For the HSC progeny, we observed a peak in contribution 8 weeks post transplantation in the Y-O group, which appeared earlier in the O-O group, namely, at 3 weeks. The contribution of CLP-derived cells was barely detectable at all time points, while a distinct contribution of CMPs was detectable at week 1 post transplantation in both experimental settings, and diminished after 3 weeks. Of note, we observed a high variability between the animals in the Y-O group compared to the O-O group, especially at later time points. A prominent example is the number of MPP-contributing cells at 16 weeks (Y-O: 7.9 ± 7.67% in comparison to O-O: 3.0 ± 1.03%). Although we measured high donor chimerism in our samples, the proportion of transduced HPSC output was comparatively low but stable. This could result from the transduction procedure itself or from a lower engraftment potential of the transduced cells after transplantation.

Next, we determined the frequency of the transduced HSPC progeny in distinct mature cell populations, namely, within the lymphoid (as assessed by their contribution to T and B cells) and the myeloid (monocytes/macrophages and granulocytes) compartments (Figure 3D, the distinct cell populations are shown in Appendix A, and the corresponding values are listed in Appendix A). In line with the previous results, the main contribution detected in the mature blood cell lineages largely came from the transduced HSCs and MPPs. For the Y-Y experiment, we detected HSC and MPP output at low levels throughout the time course in both compartments (not exceeding 10% of contribution), with higher numbers of MPP descendants in the myeloid cells. Within the lymphoid compartment, the contribution of HSCs and MPPs was barely detectable, with an exception at week 3, when we observed extraordinarily high numbers (62 ± 4.2%) of MPP descendants within the T cells. In both experiments with old recipients (Figure 3D, middle and right panel), we did not observe pronounced differences in the contribution of the transduced cells after transplantation, independently of the age of the donor cells (Appendix A). MPP-derived cells were detectable at all time points, with a consistently higher contribution to the respective compartments (myeloid or lymphoid) compared to HSCs.

The progenies of CMPs and CLPs were only detectable in the mature cell populations during the first weeks after transplantation. Surprisingly, we observed a contribution of the CLP descendants in the myeloid compartment in the Y-Y and the O-O group, corresponding to up to 30% of granulocytes in the Y-Y group. We did not detect any long-term contribution of CLPs and CMPs and therefore omitted the data from the corresponding plots (Figure 3D; 3–16 weeks post transplantation).

A remarkable difference among the old recipients was the variability between the animals. This was best documented at the last time point, 16 weeks post transplantation (compare Figure 3B,C). Whereas the values for FP-expressing cells in the O-O experiment grouped together, especially in the MPP-derived myeloid cells (Figure 3D, lower right plot), the inter-animal and cell type-specific variances in the Y-O experiment were distinctly higher (MPP in Y-O: lymphoid 5.3 ± 8.02% and myeloid 10.8 ± 9.52% in comparison to O-O: lymphoid 1.2 ± 1.13% and myeloid 9.6 ± 2.48%, Figure 3D, middle plots). To summarize, we observed that cells derived from transduced MPPs showed a higher contribution than HSC-derived cells 16 weeks post transplantation.

### 2.4. Clonal Composition of the Reconstituted Compartments

Next, we used the genetic barcode in the provirus to assess clonal aspects of the reconstitution process. To correct for multiple viral copies per cell (in the Y-O and O-O experiments), we analyzed the mean barcode numbers normalized to the VCN (VCN = 3 for the Y-O experiments and VC*N* = 2 for the O-O experiments, compare Appendix A). The number of the obtained barcodes from the Y-Y experiment was very low, as could be expected from the low initial transduction efficiency.

For the transduced HSCs and MPPs, the clone numbers (Figure 4) declined continuously over time post transplantation in the animals which received young HPSCs (Y-Y and Y-O), independently of the recipient’s age. The clone numbers stabilized towards the end of the experiment, especially for the MPP-derived cells detected in BM and spleen. For the O-O transplantations, we first observed an increase in clone numbers of HSC- and MPP-derived cells, peaking at 3 weeks post transplantation, and afterwards a decline.

A comparable pattern was observed when we studied clonal diversity in terms of the Shannon index [40,43] (Figure 5, higher values indicating a polyclonal situation with more balanced clone sizes). While the Shannon index remained largely stable for the HSC and MPP in the O-O group, the values for the Y-O and the Y-Y groups decreased more prominently over the observation period.

Analyzing the clone numbers in the progeny of transduced CMPs and CLPs, we observe that the majority of CMP-derived clones were only present in the first week, with merely a few clones detectable in the later samples (Figure 4). A similar pattern was seen in the CLP population, in which only a few clones were detectable at later time points for the Y-Y and Y-O experimental settings (Figure 5). Consistently, we obtained a different pattern for the O-O group, in which up to 400 clones were detectable in the BM and spleen at the end of the experiment.

The smaller number of detected clones also affected the clonal diversity measures (Figure 5), which were generally lower and more heterogenous for CMPs and CLPs. The increase in the detected clone numbers (compare Figure 4) at the final time point was also reflected by a corresponding increase in terms of clonal diversity.

In order to analyze the correlation between the abundance of FP-expressing cells (Figure 3) and the detected number of clones (Figure 4), we calculated the Spearman correlation coefficient for the respective experimental results (Appendix A). The strong correlation between FP expression and clone number for the O-O group confirmed the visual impression that the dynamics of reconstitution were closely connected. For the other two experiments, a comparable analysis resulted in smaller coefficients, most likely due to the high variability in the Y-O experimental setting (Figure 3B) and the low number of transduced clones in the Y-Y experiment.

Due to limitations in the available blood counts, individual clones could only be followed over time in peripheral blood samples from a subset of animals per experimental setting (Appendix A). This analysis confirmed the reproducibility of the barcode detection but did not reveal patterns of clonal expansion or extinction beyond the overall trends discussed above (Figure 4).

We conclude that both the number of contributing clones derived from transduced HSC or MPP and their clonal diversity consistently stabilized or moderately decreased over the course of the 16 weeks post transplantation. For the CMPs and CLPs, we merely detected clonal contributions on weeks 3 and 8 and an increase 16 weeks post transplantation.

## 3. Discussion

In this study, we used our multicolor barcode system to investigate the contribution of four individually cultured and transduced and, therefore, distinguishable HSPC subpopulations, namely, HSCs, MPPs, CMPs, and CLPs, after transplantation into lethally irradiated recipients. In particular, we choose three different settings (Figure 2A) to (i) investigate whether donor or recipient age affects the overall engraftment, (ii) compare the temporal contribution of the distinct subpopulations, especially in the first phase of reconstitution, and (iii) address the clonality of the engrafted cell populations emerging from the distinct transplanted subsets.

We observed that the number of HSCs sorted from the bone marrow of old donors was higher in comparison to that in the young donor material (Figure 2B) [44,45,46], potentially compensating for the functional deficit of aging HSCs. This observation is in line with previous studies [47,48,49,50]. We did not measure considerable differences in chimerism and reconstitution dynamics of the leukocyte counts between the different age groups (Figure 3).

Our experimental approach allowed us to follow the contribution of the four transduced HSPC subpopulations via their fluorescent protein expression and on a clonal level using genetic barcodes. By flow cytometry, we measured the contribution of our initially transduced HSPCs to lymphoid cells (T and B cells) and myeloid cells (granulocytes, monocytes/macrophages) (Figure 4). The main contribution to all blood cell lineages was maintained by HSCs and MPPs and appeared independent of the donor or recipient age. However, the overall level of contribution from the transduced donor cells was low, which most likely resulted from the culturing and transduction procedure. This functional impairment may lead to a limited engraftment potential and appears to level age-dependent differences. In a similar approach, Radtke et al. used lentiviral marking of different HSPC subpopulations in a non-human primate HSCT model to follow their contribution during engraftment. They found that a HSCs population consisting of CD34+CD45RA-CD90+ cells was exclusively responsible for the multilineage engraftment [51], without considerable MPP contribution. 

CMP- and CLP-derived cell populations predominantly contributed to short-term reconstitution [52,53]. The main contributions were detectable in the first days post transplantation, probably resulting from the short life span of these progenitor cells. Interestingly, we detected CLP-derived barcodes 16 weeks post transplantation in the spleen and bone marrow, especially in the O-O experiment, which were not observed in the flow cytometry data. While a retrospective characterization of these cells with respect to their phenotype is not feasible, these findings confirm the higher sensitivity of the barcoding method compared to the flow cytometry-based detection of the respective FP.

Surprisingly, we could not confirm a prominent myeloid skewing of aged HSCs, which was previously described after HSC transplantation [44,48,49,54,55,56]. We speculate that the in vitro culture for vector transduction marginalized this skewing effect. However, we detected a higher contribution to the myeloid cells for both populations (HSCs and MPPs) compared to their lymphoid contribution. This indicated that the skewing might also be present in the MPP compartment. Based on these stable contributions in our experimental approach, we reasoned that cells from the MPP pool possess a long-term reconstitution potential and can support or compensate for HSC functionality. Our results are in agreement with related findings on MPP contribution in steady-state hematopoiesis [57] and after reconstitution [58,59]. By transplanting different defined subsets of MPPs, Oguro et al. and Lui et al. found that not only HSCs, but also MPPs represent a heterogeneous cell pool that is capable of transient and long-term reconstitution [60,61]. Based on extensive immunophenotyping studies of Wilson et al. [62] and further characterization of proteomic, transcriptomic, and methylomic data [63], MPPs and their subclasses (MPP1–MPP4) were thoroughly described and functionally characterized. In our case, the MPP population was sorted based on CD150 expression, thereby resembling the MPP1 and MPP2 subclasses and providing a less lineage-restricted/differentiated form of MPPs. In our experiments, MPP-derived cells showed a higher prevalence during the reconstitution phase, both by FP expression (Figure 4) and by clonal analyses (Figure 5). 

Using the BC32 system, we could address the clonal dynamics resulting from the four cultured and transduced HSPC populations in parallel [38,40]. Although our experimental set-up allowed us to estimate temporal changes in the number of contributing clones derived from the distinct transplanted subpopulations, individual clonal time courses could only be obtained from a few animals (Appendix A). We consistently found a polyclonal pattern (Figure 4) with high clonal diversity (Figure 5) in all settings. The detected number of clones did not support the idea of only few contributing cell clones or a rapid monoclonal conversion in any of the three experimental settings. The contribution of the more differentiated populations met our expectations, as we observed a rapid decrease in CMP-derived clone numbers within the first weeks, or even the absence of any CLP-driven contributions. Since we did not analyze the presence of CLP-derived clones in the thymus, the observed occurrence of CLP derivatives after 16 weeks might be explained by the presence of newly generated mature T cells (Figure 5, bottom panel).

### Limitations

Some aspects of the experimental design limit the potential conclusions of our study. Although we applied an equal ex vivo manipulation protocol of the cells, e.g., sorting of the respective populations, cultivation and transduction protocols of HSPCs, the transduction efficiency of the Y-Y group was far lower compared to that of the other two experimental groups. While we compensated for this effect by normalization on the vector copy number, an unbalanced fraction of transduced cells between the three groups remained. In gene therapy studies, it was shown that non-transduced cells had a higher engraftment potential than transduced cells [64], which might also explain the lower fraction of transduced cells in our recipient animals. Furthermore, the assessment of hematopoietic organs at several time points during follow-up did not allow tracking single clones during the whole observation procedure within the same animal. However, we aimed to follow the overall clonal dynamics especially in the first phase of reconstitution, in which only limited cell numbers were detectable within peripheral blood and bone marrow. This led us to the decision to measure the clonal composition in total peripheral blood, total bone marrow, and total spleen from the animals, without previously sorting for FP-positive cells or mature blood cells. This broader approach may have shaded the underlying clonal dynamics of the transduced cell population, as smaller clones might have been missed due to the sampling of the DNA. Age-mismatched transplantations with old donor material transplanted into young recipients (i.e., O-Y) would further broaden the spectrum of our analysis and are of special interest for future characterizations of aged HSPC populations.

## 4. Materials and Methods

### 4.1. Generation and Production of Barcoded Lentiviral Vectors

The generation of vector constructs and their subsequent equipment with the BC32 barcoded plasmid library (Figure 1A), were described by Aranyossy et al. [40]; the utilized oligos are shown in Appendix A. VSV-g pseudotyped lentiviral particles were produced as described [65].

### 4.2. Isolation and Transduction of the Target Cells

Animal experiments were performed in accordance with ARRIVE guidelines and legal regulations after approval by the Institutional Animal Care and Use Committee of the City of Hamburg (Hamburg 89/14) [66]. Mice were bred in-house in the animal facility of the University Medical Center Hamburg-Eppendorf (UKE).

Thirty male C57Bl/6 mice were used as donors for each experimental setting. Cell isolation from the BM of the donors (young: 2 months of age; old: 18 months of age) was performed as described by Aranyossy et al. [40]. After staining with an established antibody panel, based on lineage depletion and the markers Sca-1, ckit, and CD150 (Appendix A) [14,60,62,63,67], the cells were sorted using BD FACS Aria IIIu (BD Biosciences, Franklin Lakes, NJ, USA) into HSCs, MPPs, CMPs, and CLPs (Appendix A) and subsequently transduced with the respective barcoded vector in StemSpan medium containing 50 µg/µL Stem Cell Factor (SCF, PeproTech, London, UK). The cells were combined into grafts on the next day or underwent fluorescent protein (FP) measurement via flow cytometry 3 days post transduction. As an alternative method to estimate the transduction efficiency, we fitted a Poisson distribution to barcode numbers from single cells (Appendix A).

### 4.3. Murine Transplantation Experiment

The four transduced subpopulations were combined and mixed with BM cells of females age-matched to the recipients (1 × 10^6^ cells per recipient), to ensure the engraftment and the survival of the mice. The cell mixture was divided by 30, and each graft was transplanted into one of 30 lethally irradiated (9.5 Gy) females by tail-vein injection. To follow the reconstitution, groups of five to seven animals were analyzed at once, 1, 3, 8, and 16 weeks post transplantation. To follow the clonal kinetics within the same individual, intermediate blood samples were taken from the last group (final analysis at 16 weeks post transplantation). Single-cell suspensions from PB, BM, and spleen were analyzed by flow cytometry and next-generation sequencing (NGS) (Figure 1B). The cells from the spleen were additionally stained for CD3, B220, CD11b, and Ly6G to determine the HSPC progeny in the T cell, B cell, granulocyte, and monocyte/macrophage compartments (Appendix A). For VCN determination, single cells of HSC or MPP origin were sorted according to their FP expression from the spleens of different animals. Up to 21 individual cells were selected per FP color, and the barcode sequences were PCR-amplified. To estimate the VCN, we analyzed the Sanger sequencing chromatograms (Appendix A) [68].

### 4.4. Determination of Chimerism

We performed ddPCR on DNA from the BM in a duplex reaction by simultaneously amplifying the Y chromosome and a reference gene (erythropoietin receptor). The analyses were performed using QX100 (BioRad, Hercules, CA, USA) according to the manufacturer’s instructions, with the primer and probes shown in Appendix A [38,42,69].

### 4.5. Barcode Analyses

The genetic barcodes were extracted from gDNA and amplified via PCR as described earlier [39,40]. In brief, 200 ng of genomic DNA from PB, BM, or spleen was used in a single PCR reaction, using the oligos listed in Appendix A. The obtained PCR products were purified using Agencourt XP-beads (Beckman Coulter, Brea, CA, USA) and quantified using the Qubit System (Life Technologies, Carlsbad, CA, USA). Finally, up to 65 PCR products were mixed in equal amounts to compose the libraries, which were subsequently sequenced on the Illumina MiSeq System (Illumina, San Diego, CA, USA) using single-end reads of 83 bps length. After demultiplexing, the NGS results were analyzed utilizing an in-house developed R-package [39,70]. Barcode counts were corrected for the VCN (Appendix A), reflecting the reconstitution of the HSPC-derived clones. Subsequent measurements and clonal tracing within the same recipient were performed in animals from the last group (16–18 weeks) (Appendix A). We used the R-Package “vegan” to calculate the Shannon index of our samples (https://cran.r-project.org/web/packages/vegan/index.html, last access 28 January 2022).

## 5. Conclusions

In summary, our approach documents the feasibility of individual marking and parallel transplantation of four distinct HSPC subpopulations to follow their contribution to reconstitution after HSCT. In our murine transplantation experiments, we focused on the first reconstitution phase up to 16 weeks post transplantation, in which we could not see any profound impact of donor or recipient age. We detected the same kinetics as described for HSCTs in murine or human CD34 xenograft experiments [18,30] or in transplantation of macaques [36], with high cell numbers in the beginning, most likely derived from less potent progenitors (here: CMPs), and a stable contribution of multipotent clones in the longer run that were derived from the MPPs and HSCs. Our data suggest that donor and recipient age were not influential, most likely as a result of the culture and transduction procedures, and did not affect the overall reconstitution dynamics after transplantation. We consistently observed a higher contribution of MPPs over HSC-derived output, hinting towards a partial impairment of the HSCs during transduction. Stable levels of clonal diversity in HSCs and MPPs did not suggest an accelerated clonal dominance in any of the aged settings. As expected, the CMP- and CLP-derived output was only detectable in short-term reconstitution, confirming the limited capability of these progenitor cells to contribute to long-term reconstitution.

Our experimental approach closely mimics an autologous transplantation setting in which the transduced cells need to achieve a sustained contribution in the particular recipient. Such settings become increasingly important in the context of gene therapeutic applications [71,72]. Newer protocols, e.g., stem cell gene therapy for sickle cell disease, will also include adult patients [73] and provide a potential reference system in the future. Especially for the clinical setting, it is of greatest interest to analyze the differential susceptibility to lentiviral transduction of different HSPC populations within the heterogeneous CD34+ cell pool and to elucidate the impact of genetically modified subpopulations on the success of transplantation in stem cell gene therapy approaches. Until now, the composition of the CD34+ cell population in patients’ grafts has not been further examined with regard to the frequency of bona fide HSCs and MPPs. Our results suggest that both populations play an important role in sustained multi-lineage engraftment, whereas donor or recipient age might be less of a limiting factor for cultured and transduced graft cells.

## Figures and Tables

**Figure 1 ijms-23-03160-f001:**
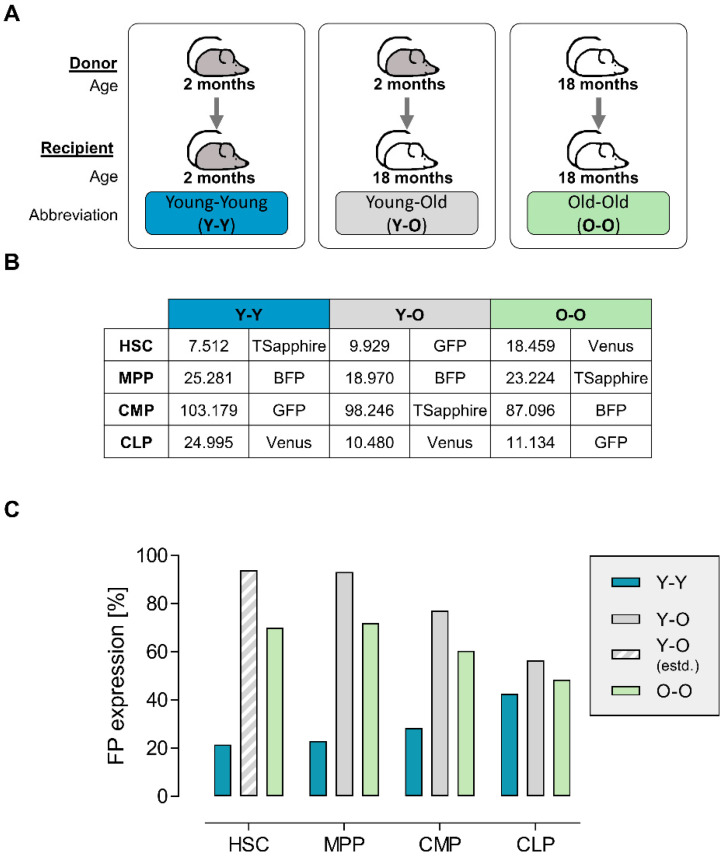
Experimental groups, sorted cell numbers, and transduction efficiencies. (**A**) The experimental groups were set up as follows: cells from 2-month-old male mice were used for transplantation into 2-month-old female recipients (blue: Y-Y), cells taken from young donors (18 months old) were transplanted into old recipients (2 months old, grey: Y-O), and finally cells from old donors were transplanted into old (both 18 months old) recipients (light green: O-O). (**B**) According to the marker profile, the cells were separated by FACS and transduced with lentiviral barcode vectors encoding the depicted fluorescent protein. Numbers depict the mean number of cells per animal obtained for each subpopulation after sorting. (**C**) Three days post transduction, transduction efficiencies for the four subpopulations were measured by flow cytometry of the respective fluorescent proteins. The value for the transduced HSCs in the Y-O experiment was estimated using Poisson distribution (grey striped bar). Blue bars represent the Y-Y experimental setting, grey bars represent the Y-O experimental setting, and light green bars represent the O-O experimental setting.

**Figure 2 ijms-23-03160-f002:**
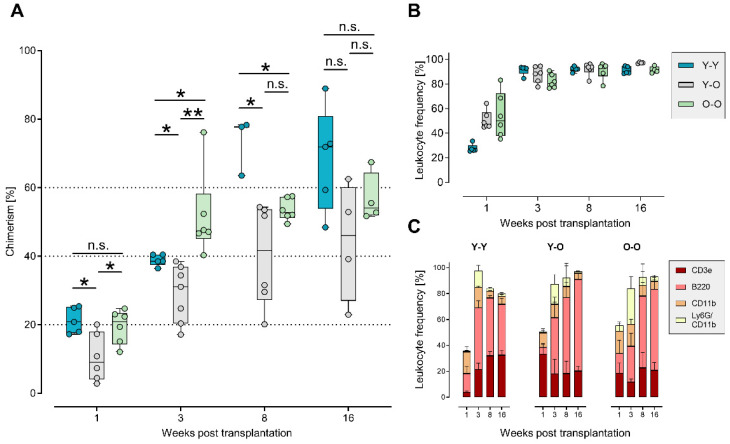
Reconstitution dynamics. (**A**) We determined the chimerism via a Y chromosome-specific ddPCR in the bone marrow of transplanted animals. The pair-wised Mann–Whitney U-Test showed some statistical significance in reconstitution dynamics between the three groups, despite great inter-animal variances. Non-adjusted p-values are indicated as not significant (n.s.) for *p* > 0.05, * for *p* < 0.05, and ** for *p* < 0.01. Blue bars represent the Y-Y experimental setting (*n* = 3–5), grey bars represent the Y-O experimental setting (*n* = 4–7), and light green bars the O-O experimental setting (*n* = 4–6). (**B**,**C**) We followed leukocyte reconstitution in the spleen for all leukocytes (in **B**) and specifically for the respective subpopulations, i.e., T cells (CD3e), B cells (B220), monocytes/macrophages (CD11b), and granulocytes (CD11b and Ly6G) (in **C**). The frequencies of these populations after transplantation were similar, independent of the age of the donor mice (Y-Y and Y-O vs. O-O).

**Figure 3 ijms-23-03160-f003:**
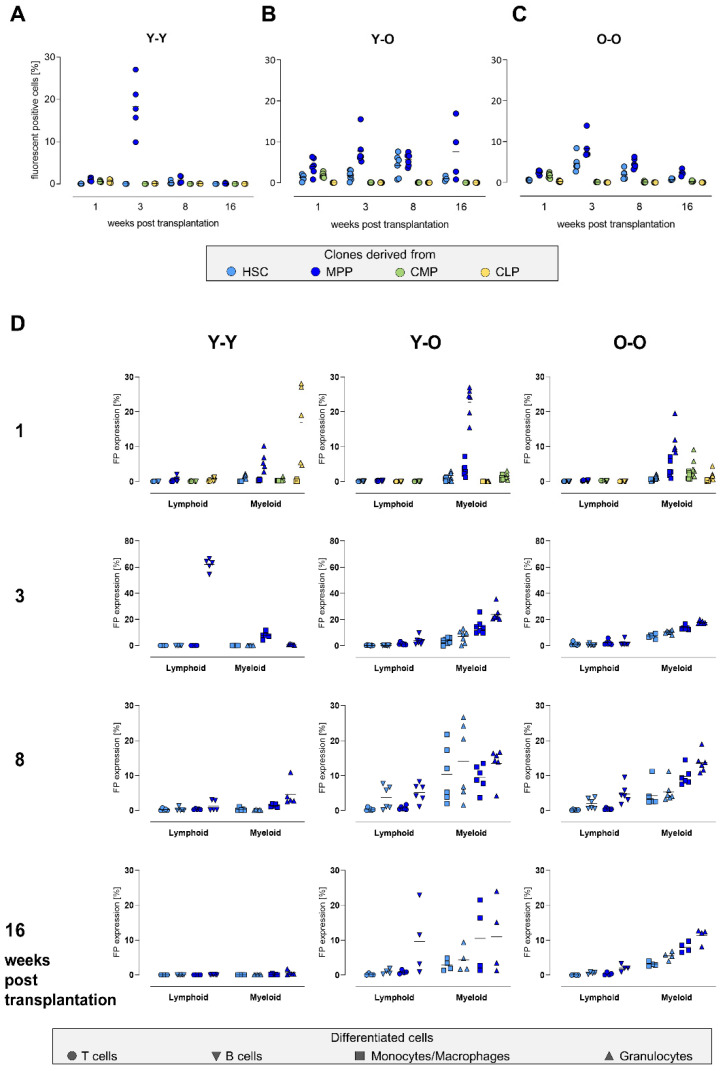
Contribution of the transduced cell populations. (**A**–**C**) Single-cell suspensions from the spleen were used to determine the overall contribution of the initially marked cell populations via their fluorescent protein expression by flow cytometry. (**D**) FP expression was also measured in the lymphoid (T cell (CD3) and B cells (B220)) and myeloid (monocytes/macrophages (CD11b) and granulocytes (CD11b/Ly6G)) compartments to analyze their contribution in more detail. The spleen mostly comprised HSC- and MPP-derived cells. CMP-derived cells were solely detectable 1 week after transplantation. CLP-derived cells were barely detectable. No data were available for CMP- and CLP-derived cells 3, 8, or 16 weeks post transplantation. Light-blue dots: HSC-derived clones, dark-blue dots: MPP-derived clones, green dots: CMP-derived clones, and yellow dots: CLP-derived clones.

**Figure 4 ijms-23-03160-f004:**
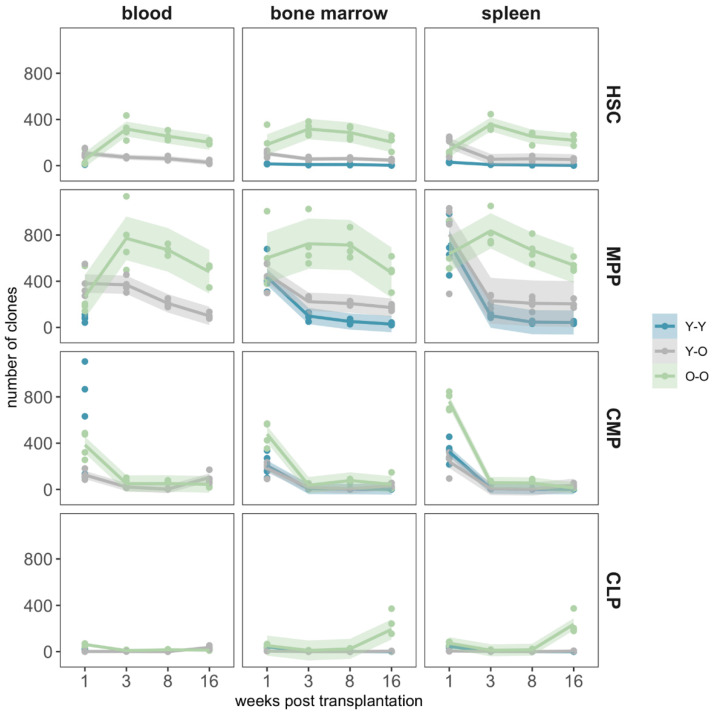
Mean barcode content. We extracted the DNA from peripheral blood, bone marrow, and spleen and amplified the BC32 sequences by PCR. NGS and subsequent bioinformatic analyses showed the number of unique barcodes from the initially marked cell population in the respective organ, corrected by the vector copy number (VCN). For a better representation of the reconstitution dynamics, we used fitted local polynomial regression curves (lines) including their respective 95% confidence bands (shaded regions) to link the individual clone numbers (points). Blue data represent the Y-Y experimental setting, grey data represent the Y-O experimental setting, and light green data represent the O-O experimental setting. The data points from the Y-Y setting are mostly overlayed by the others.

**Figure 5 ijms-23-03160-f005:**
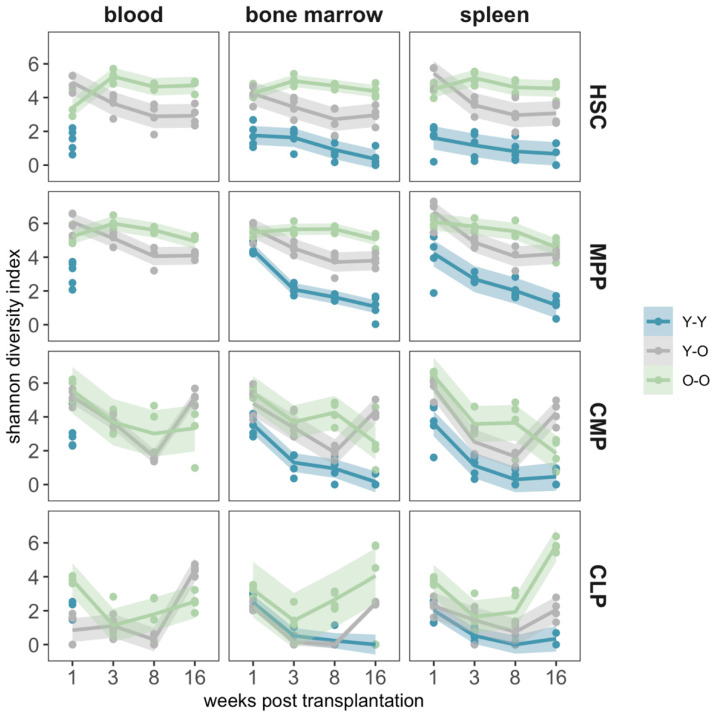
Clonal diversity measured by the Shannon index. We calculated the Shannon index (36, 48) to determine the clonal diversity in peripheral blood, bone marrow, and spleen. A high Shannon index refers to a polyclonal situation, whereas a lower index reflects a loss of clones or the emergence of clonal dominance. A fitted local polynomial regression curve (including a 95% confidence interval) was used for a better representation. Blue data represent the Y-Y experimental setting, grey data represent the Y-O experimental setting, and light green data represent the O-O experimental setting.

## Data Availability

The data that support the findings of this study and any custom written code are available from the corresponding author upon reasonable request.

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
