# Peer review of "The Reconstitution Dynamics of Cultivated Hematopoietic Stem Cells and Progenitors Is Independent of Age"

_ijms, 2022, doi:10.3390/ijms23063160_

Round 1
Reviewer 1 Report
The authors have performed a huge amount of work to determine whether the age of the donor and recipient influence the transplantation output using some novel barcoding technologies developed by the same authors. The authors conclude based on their analysis that no differences are observed in young versus aged donors/recipients in terms of output.
The results presented here seem to go “against” several publications that claim that aged mice contribute less to reconstitution. Although it seems counterintuitive, the reviewer thinks that the “no differences” seen in this study might be easily explained by the fact that the authors have to cultivate and transduce cells in vitro. Therefore, it would be wrong to conclude that these cells are not performing differently (which is seen when directly transplanting freshly isolated cells) but rather, and also very excitingly, that cells after cultivation do not differentially contribute to the reconstitution, which is per se an important information for the scientific community since most of the cells transplanted in patients are being cultured before transplantation. Thus, the reviewer strongly suggest the authors to twist the angle of the here presented story and emphasize in the title, abstract and discussion the word “cultivated” and the fact that HSPCs get activated and differentiate in vitro and as a consequence, young and old do not anymore performed functionally different. In addition, since the transduction system has per se some limitations, I also strongly suggest to “tone down” the conclusions and use rather the word “suggest” and also the title (e.g. using the word cultured,.) Also in the abstract “Our experiments show that the dynamics of reconstitution and the contribution of individually transduced HSPC subpopulations is largely independent of age.” should be toned down.
The authors have also nicely discussed potential pitfalls of their strategy (particularly since they had to used slight different strategies at the beginning due to availability of cell numbers)
The authors should also discuss the work of Camargo and Rodewald groups on in vivo barcoding.
Minor suggestions:
Typos in Figure A1A I guess here the authors meant Figure 1A.
Reviewer 2 Report
Gotzhein and others describe the influence of donors and recipients age on the reconstitution kinetics of different hematological subpopulations. They showed nicely that the reconstitution patterns were independent of age, which could be a strong rationale for application of HSCT in older patients. Considering aging of the general population this conclusion is of major importance.
Major Comments:
- Study designe. In my opinion, one more pattern of donor-recipient model should be also presented in the paper, which is old to young (O-Y) pattern. These results might be also interesting, in the terms if also the age did also not influence hematological reconstitution in this setting.
Minor Comments:
- Please unify the writing “Supplementary Table …” or “Supplementary table …”, p 13.
- Supplementary materials: p 5-6, Supplementary Figure 4and Figure 5, should be written.
